# Advances in Tumor-Infiltrating Lymphocyte (TIL) as a Prognostic Factor and for Treating Invasive Cutaneous Melanoma

**DOI:** 10.3390/ijms252312596

**Published:** 2024-11-23

**Authors:** Gabriel Alves Freiria de Oliveira, Daniel Arcuschin de Oliveira, Melissa Yoshimi Sakamoto Maeda Nisimoto, Rafael Rubinho, Heitor Carvalho Gomes, Luciana Cavalheiro Marti, Renato Santos de Oliveira Filho

**Affiliations:** 1Melanoma and Skin Tumors Sector, Plastic Surgery Discipline, Escola Paulista de Medicina/Universidade Federal de São Paulo (EPM/UNIFESP), Rua Botucatu, 740, São Paulo 04023-062, SP, Brazil; gafoliveira@unifesp.br (G.A.F.d.O.); arcusha@gmail.com (D.A.d.O.); mel_maeda@yahoo.com.br (M.Y.S.M.N.); heitorgomesmd@uol.com.br (H.C.G.); 2Department of Dermatology, Universidade Santo Amaro, Rua Isabel Schmidt, 349, Santo Amaro, São Paulo 04743-030, SP, Brazil; rafael_rubinho20@hotmail.com; 3Department of Experimental Research, Hospital Israelita Albert Einstein, Rua Comendador Elias Jafet, 755, Morumbi, São Paulo 05653-000, SP, Brazil

**Keywords:** melanoma, metastatic melanoma, tumor-infiltrating lymphocytes (TILs), cutaneous melanoma

## Abstract

Invasive cutaneous melanoma is responsible for about 5% of skin tumors yet is liable for nearly 70% of skin cancer-related deaths. Despite notable advancements over the past decade, including immunotherapies and targeted treatments, more than half of invasive melanoma patients ultimately succumb to the disease due to therapeutic resistance. To overcome this obstacle, strategies such as combining immunotherapies with targeted drugs or adding epigenetic therapies have been investigated. Tumor-infiltrating lymphocytes (TILs) therapy has emerged as a promising option for patients whose disease continues to progress despite standard treatments. This article aims to introduce TIL therapy and review recent outcomes in melanoma prognosis in its application for melanoma management.

## 1. Introduction

Invasive cutaneous melanoma is responsible for around 5% of skin cancer cases, yet it is highly lethal, accountable for over 70% of skin cancer-related deaths [1]. The worldwide incidence of melanoma in 2020 was 324,635 cases, with a total of 57,043 deaths [2]. In the United States, it is projected that 200,340 melanoma cases will be diagnosed in 2024. Of these, 99,700 cases are expected to be noninvasive (in situ), while 100,640 will be invasive. Among the invasive cases, 59,170 will emerge in men and 41,470 in women [3].

The prevalence of melanoma varies according to region and differs between countries. This fact is attributed to variations in skin phenotypes, as well as differences in sun exposure. Unlike other solid tumors, cutaneous melanoma mainly affects young individuals [2]. In Brazil, where much of the territory lies in intertropical regions with high solar incidence, non-melanoma skin cancer is the most prevalent form of cancer, representing around 30% of all malignant tumors registered in the country; melanoma represents 3% of malignant neoplasms of the organ. Furthermore, the number of new cases is expected to rise in the coming years, with estimates suggesting more than 220,000 new cases of skin cancer between 2023 and 2026 [4].

Despite the progressive increase in melanoma incidence, five-year survival rates have significantly improved due to early diagnosis and advances in treatment with targeted therapies, such as B-Raf Proto-Oncogene (BRAF) and Mitogen-Activated Protein Kinase Kinase (MEK) inhibitors, and immunotherapies involving immune checkpoint inhibitors like anti-Programed Cell Death Protein 1 (PD-1), anti-Programed Cell Death Protein Ligand-1 (PD-L1), anti-Cytotoxic T-Lymphocyte-Associated Protein 4 (CTLA-4), and anti- Lymphocyte-Activation Gene 3 (LAG-3). Some examples of these drugs are Vemurafenib, Dabrafenibe (BRAF), Cobimetinib (MEK), Nivolumabe, Espartalizumabe, Pembrolizumabe e Toripalimabe (PD-1), Atezolizumab (PD-L1), Ipilimumabe (CTLA-4), and Opdualag (LAG-3) [5,6]. While these therapies have shown success, some patients do not respond initially, and others develop resistance over time [7]. The immune system has been extensively studied in the context of cancer, and various strategies to stimulate it to fight cancer have been developed. One approach is the use of tumor-infiltrating lymphocytes (TILs), which will be the focus of this review.

The first immunotherapy treatment using interleukin-2 (IL-2) was introduced by Steven Rosenberg in the 1980s; this therapy demonstrated effectiveness in shrinking tumors and enhancing patient outcomes. This breakthrough has led to significantly prolonged survival and enhanced quality of life for many cancer patients. Rosenberg’s pioneering work paved the way for the development of various immunotherapies, which have been instrumental in combating previously incurable cancers [8]. Today, high-dose interleukin (IL)-2 therapy, although associated with significant side effects, remains a critical treatment for metastatic melanoma and renal cell carcinoma, offering durable responses in a subset of patients [9]. Rosenberg continued to spearhead basic and clinical research, leading to the development of adoptive cell therapy in 1988. This groundbreaking treatment involved extracting T lymphocytes with antitumor activity from a cancer patient, expanding their numbers through laboratory cultivation, and then re-infusing them into the patient to enhance the immune response against cancer [8].

TILs were defined as a heterogeneous population of lymphoid cells, which include CD4+ helper T (Th) cells and CD8 cytotoxic T cells (Tc). Among the Th cells, Th1 supports anti-tumor immunity by secreting IFNγ, while regulatory T cells (Treg), can suppress immune responses and promote tumor tolerance. Additionally, subsets like Th17 cells may influence inflammation within the tumor, and follicular helper T (Tfh) cells can help the B cells responses. Tc are the cytotoxic cells that can kill the tumor cells through degranulation of granzymes and perforin. Since these T cells are infiltrated in tumor microenvironments, they are also supposed to be most able to recognize tumor antigens (tumor-specific T cells) [10]. Furthermore, the interaction between TILs and tumor cells is complex and context-dependent; tumor signaling can suppress the immune response, promoting a regulatory environment (Figure 1A). However, the immune system retains the ability to recognize and eliminate tumor cells (Figure 1B).

The process of obtaining and applying TILs involves several steps. First, lymphocytes naturally infiltrating tumor tissues are isolated from biopsies of patients with metastatic cutaneous melanoma. These lymphocytes are then cultivated and expanded in the laboratory intending to expand and activate the Tc and Th1 phenotypes in tumor antigen-specific that are infiltrating the tumor, as depicted in Figure 2. A large quantity of the expanded cells, along with IL-2 in high doses, is re-infused in the patient. Prior to TIL infusion, patients undergo a nonmyeloablative lymphodepletion regimen.

Pioneering work with TILs was conducted by Dr. Steven Rosenberg’s research group in 1986. They isolated TILs from established murine tumors, including the B16 and M3 melanomas and amplified them in vitro using IL-2. These TILs were then re-infused in tumor-carrying hosts, resulting in a substantial antitumor response. Additionally, they discovered that combining IL-2 with TIL infusion could enhance the therapy effect, mediating the regression of pulmonary and hepatic metastases from these tumors. Following this, Rosenberg applied TIL therapy to melanoma patients, and its successful outcomes were published in 1988 [11,12,13].

Today, several cancer institutes worldwide have applied TIL therapy, yielding excellent results in patients with invasive melanoma [14,15]. In this article, we present the importance of TILs as a prognostic factor and an overview of the current status of TIL therapy in invasive cutaneous melanoma, covering the TILs preparation process, treatment protocols, the application of combined therapies, and innovations that can add therapeutic value to this protocol.

## 2. Results

(a)Prognostic value of TILs in primary cutaneous melanoma

Alexandru Gata et al. evaluated the significance of TILs as a prognostic factor for metastatic lymph node and survival in patients with fully resected stage pT3 melanoma. The study enrolled 114 patients treated between 2000 and 2015, and all the patients presented pathological pT3 cutaneous melanoma. This study analyzed the relationships between clinical and disease factors with lymph node status and patients’ survival rates. The results indicated a rapid TIL infiltration in 60% of patients, while 40% showed either absent or slow infiltration. Univariate analysis revealed a correlation between the presence of ulceration and absent TILs. In multivariate analysis, lymph node invasion and absent TILs were associated with a higher risk of death. Thus, the density of TILs serves as an independent prognostic factor for overall survival, providing an accurate prognosis for patients with pT3 melanoma [12].

A study by Angeramo et al. highlighted the significance of tumor-infiltrating lymphocytes (TILs) in melanoma survival (MS), relapse-free survival (RFS), and sentinel lymph node (SLN) status. They conducted a retrospective analysis of patients who underwent melanoma resection between 2009 and 2019. Using the Melanoma Institute Australia (MIA) grade system for TILs, the participants were separated into two groups: G1 with TIL grades 1, 2, or 3, and G2 with TIL grade 0. Out of 386 melanoma excisions, 151 (39%) were categorized as G1, and 39 (10%) as G2. Amongst the patients who received SLN biopsy, the frequency of results based on TIL grades 0, 1, 2, and 3 was 32%, 18%, 14%, and 0%, respectively. Patients were followed for 48 months, and 5-year melanoma-specific survival rates were significantly higher in G1 (86%) compared to G2 (75%). Similarly, 5-year recurrence-free survival rates were markedly higher in G1 (81%) versus G2 (60%). This study highlights the association between TIL and SLN status with improved melanoma-specific survival (MS) and recurrence-free survival (RFS) [13].

Klein et al. assessed the prognostic relevance of TIL clusters in primary metastatic melanomas (MM) and examined their association with the disease’s molecular subtypes to predict the response to immune checkpoint inhibitors (CPI). They analyzed 90 MM patients treated with checkpoint inhibitors (CPI) and a validation cohort of 351 patients from the TCGA (SKCM) database who received conventional treatment. For image segmentation, this study utilized a U-Net deep convolutional neural network (a network and training strategy that relies on the strong use of data augmentation to efficiently utilize the available annotated samples) to detect viable tumor regions in whole-slide images, while a separate neural network quantified TIL numbers. The count of TIL clusters correlated with CPI response in 90 MM patients, with a stronger association observed in BRAF V600E/K-mutated cases. However, TIL cluster numbers in NRAS-mutated and wildtype (BRAF-wt, NRAS-wt) MM tumors showed no predictive value for CPI response. PD-L1 expression had restricted prognostic relevance in this cohort. In a separate group of MM patients from the TCGA cohort (n = 351), the number of TIL clusters was associated with improved survival in BRAF V600E/K-mutated cases, but not in NRAS-mutated or BRAF/NRAS-wildtype MM patients. While in MM, TILs have been associated with improved survival, and specifically, TIL clusters have shown a correlation with immunotherapy response in BRAF V600E/K-mutated cases [14].

Tas and Erturk investigated the potential roles of histological regression (HR) and tumor-infiltrating lymphocytes (TILs), both individually and in combination, as early indicators of immune activation against primary melanoma and their impact on survival. In this study, 916 cutaneous melanomas were retrospectively analyzed. Histological regression (HR) was observed in a minority of lesions (25.1%) and was correlated with male sex, axial location, non-nodular histopathology, thin Breslow depth, and nevi-associated melanoma. TILs were present in 48.4% of cases and were correlated with lower Clark levels, thin tumor thicknesses, lower mitotic rates, BRAF mutation, absence of neurotropism, lymph node involvement, and disease recurrence. A significant correlation was found between HR and TILs. TILs were associated with favorable recurrence-free survival, while no connection was observed between HR and recurrence-free survival [16].

The study of Zablocka et al. compared the value of assessing TILs in I and II stages of nodular melanoma (n = 56) and superficial spreading melanoma (n = 29), along with BRAF mutational status and their associations with clinicopathological characteristics. Perilesional lymphocytes were defined as lymphocytes around the tumor mass. Their dispersion was scored as absent (0), covering less than 25% of the tissue (1), covering between 25% to 50% of the tissue (2), and covering more than 50% of the tissue (3). Low TIL infiltration was classified as scores of 0 and 1, while high TIL infiltration was classified as scores of 2 and 3. The results corroborate a recent study confirming that the coincidence of tumor regression and TILs is related with prolonged survival in melanoma. However, this study did not demonstrate an association between tumor regression and TILs, which may be due to the inclusion of only stage IA–IIC patients. Nodular melanomas showed a decreased presence of TILs compared to superficially spreading melanomas, potentially due to intratumorally heterogeneity, varying cytokine environments, and gene expression. This observation was particularly noted in thin melanomas (≤2 mm) and male patients, suggesting distinctions in immunological response based on gender and tumor depth. Additionally, tumor stage and the TIL heterogeneity could have contributed to these findings [17].

The multicentric study by Morrison et al. evaluated the association of TILs with melanoma regression. Data from the Sentinel Lymph Node Working Group database (1993 to 2018) were queried for TIL and melanoma regression information. Clinicopathologic factors were correlated with regression, TIL status, sentinel lymph node (SLN) status, and overall survival (OS). The study enrolled 2450 patients, with TILs present in 1811 cases (73.9%). Melanoma regression was observed in 328 patients, representing 18.1% of cases with TILs, while only 7.7% of the 639 cases without TILs showed melanoma regression. The presence of TILs was significantly associated with melanoma regression and a negative status for SLN. Once TILs were stratified by regression status, only the combined presence or absence of both TILs and regression was significantly associated with SLN metastases. While TILs were associated with OS, regression alone was not. When stratified by regression status, only the presence of TILs, with or without regression, was significantly connected with improved OS, whereas regression alone had no significant impact on OS. This suggests that, although regression is strongly correlated with TILs, only TILs are significantly associated with SLN metastasis and OS, indicating that the effect of regression on outcomes depends on the presence of TILs [18].

Aung et al.’s study analyzed data from 785 patients across five separate cohorts from various institutions to validate the automated TIL enumeration in order to define a subset of melanoma patients with low risk for disease recurrence after surgical treatment. Using serial tissue sections, conventional and multiplex fluorescent staining were performed to characterize TIL phenotypes and their associations with survival outcomes. Five TIL variables, with the calculated proportion of TILs relative to tumor cells (eTILs), or other cell types/areas within the tissue section (etTILs, esTILs, eaTILs and easTILS) each showed significant associations with overall survival. Additionally, they demonstrated that eTILs and etTILs (TILs over total number of cells) grades are strong prognostic indicators in patients with primary melanoma. In addition, it may help identify a subset of stage II patients at high risk of relapse who could benefit from adjuvant therapy. They also displayed a molecular correlation behind these scores. These data confirmed the necessity of prospective evaluation of these algorithms in a clinical trial [19].

Straker III et al. studied 1017 patients with cutaneous melanoma ≥1.0 mm thick who underwent excision surgery and sentinel lymph node (SLN) biopsy at a single institution from 2006 to 2019. Patients were graded by TIL status in the primary tumor, as brisk (bTILs), non-brisk (nbTILs), or absent (aTILs). Relationships between patient variables and outcomes were investigated by multivariable analysis. Of the 1017 patients evaluated, 846 (83.2%) had primary TILs: non-brisk TILs (nbTILs) in 759 patients (89.7%) and brisk TILs (bTILs) in 87 patients (10.3%). In contrast to those without TILs, patients with any type of TILs displayed higher rates of regression and lower rates of acral lentiginous histology, with rates of 84.0%, 71.8%, and 68.4% for bTILs, nbTILs, and aTILs, respectively. For the 114 immune checkpoint blockade (ICB) naïve patients who relapse after ICB therapy, no association was seen between progression-free survival and bTILs or nbTILs. Among patients undergoing excision surgery and SLN biopsy for intermediate to thick primary cutaneous melanoma, primary TILs were associated with sentinel lymph node status and improved recurrence-free survival, but not with disease-specific survival. Moreover, primary TILs did not provide prognostic value regarding responsiveness to immune checkpoint blockade (ICB) therapy for disease recurrence [20].

Collectively, these studies underscore the critical role of TILs as prognostic and predictive biomarkers in melanoma, contributing to the evolving landscape of melanoma treatment and personalized immunotherapy approach.

(b)TILs in the treatment of invasive cutaneous melanoma

Van den Berg et al. conducted a phase I/II study on TIL therapy in metastatic melanoma at the Netherlands Cancer Institute to evaluate its viability and potential for a randomized phase III trial. Ten patients were included in the trial for TIL therapy. The TILs and peripheral blood samples were characterized for phenotype and neoantigen responsiveness. The study showed lasting clinical outcomes and translational data on neoantigen responsiveness of TILs. Out of a total of ten patients, half of those who were anti-PD-1 naïve showed an objective clinical response, including two patients who have maintained complete responses for more than seven years. Immune assessment revealed detectable neoantigen-specific T cells in TIL infusions in all three patients analyzed, with significant increases in T cell response level in the peripheral blood post-therapy. Neoantigen-specific T cells persisted measurably for up to three years after TIL infusion. The clinical results reinforce the strength of TIL therapy in metastatic melanoma and emphasize the potential role of neoantigen-specific T cell responsiveness, supporting the initiation of a multicenter phase III TIL trial [11].

Sarnaik et al. conducted a phase II open-label, single-arm, multicenter study including 73 patients with invasive melanoma formerly treated with checkpoint inhibitors and/or target therapy with BRAF and MEK agents. Lifileucel was generated from harvested tumor sample TILs using an efficient 22-day entire process. The protocol included a nonmyeloablative lymphodepletion regimen, followed by lifileucel single infusion, and up to six high doses of interleukin (IL)-2. An objective response was achieved in 31.5% of patients (RECIST 1.1), with a median response duration of 18.6 months and 43.5% of responses lasting over 12 months. The median overall survival was 17.4 months, with a one-year of 38% stable disease overall survival rate and 92% for those with an objective response. The safety report was coherent with the known adverse events related to lymphodepletion and IL-2. This treatment demonstrated lasting responses, addressing a significant unaccomplished demand for metastatic melanoma patients with restricted treatment options after acquiring resistance to the approved therapies, including the ones refractory to anti-PD-1 or PD-L1 therapy [15].

Kristensen et al. investigated how the variety, frequency, and persistence of antigen-specific CD8 T cells in TIL products influence the outcomes of patients receiving adoptive T cell therapy (TIL-ACT). They mapped CD8 T cells specific for tumor antigens in TIL products and blood samples from 26 metastatic melanoma patients. They have identified 106 types of antigen-specific T cells within products, indicating a 1.8% response against all predicted tumor antigens. Antigen-specific T cells were practically absent in the TIL products infused into progressive-disease patients. The rate of antigen-specific CD8 T cells in TIL products was associated with patients’ increased survival. Additionally, the antigen-specific CD8 T cells present in the post-treatment patients’ blood samples were characteristic of those of responders to TIL-ACT. Lastly, the transcriptional signature which indicates T cell activity within the tumor microenvironment was correlated with a higher frequency of antigen-specific CD8 T cells in the infusion product [21].

Julve and Furness reviewed tumor-infiltrating lymphocyte (TIL) therapy and showed its promising activity in patients with cutaneous melanoma refractory to immune checkpoint inhibition. Activated T cells can be immune regulated, which is an evident justification for combining TIL therapy with ICIs (immune checkpoint inhibitory antibodies); not only to favor the activity of infused TILs, but also to potentiate endogenous T cells. Several studies (NCT04165967, NCT03475134, NCT03374839, NCT01701674, NCT03638375, and NCT02652455) are pursuing the impact of combined approaches, including IOV-COM-202 (NCT03645928); this is a multicenter phase II study that is evaluating TILs in combination with pembrolizumab (pembro) in melanoma patients naïve of ICI, head and neck cancer (HNSCC), and non-small cell lung cancer (NSCLC). The patients included in this study receive a single dose of pembro (after tumor resection and prior to lymphodepletion) as well as adjuvant IL-2. According to the study C-144-01, pembro is maintained after TIL infusion, six times a week, for up to 24 months. Preliminary results were presented in 2021 at an annual meeting, and even though only 10 patients were included in the melanoma group, they observed an important response rate of 60% (6/10). The efficacy-evaluated patients displayed tumor burden reduction, and three patients reached complete response. Safety was considered according to the known profile of non-myeloablative lymphodepletion, IL-2, and pembro therapies [22].

Betof Warner et al. reviewed the development of TIL therapy for melanoma and other solid tumors, highlighting past and present experiences, challenges, practical considerations, and future aspirations. Modern TIL therapy has seen advancements in manufacturing, efficiency, therapeutic protocols, and product characterization. Lifileucel, an autologous and not modified TIL infusion, has shown promising results in patients with stage III and IV unresectable melanoma after PD-1 therapy. In a phase II, multicenter, international, single-arm, multicohort study, TILs were produced mainly within three weeks without modification. One cohort of 66 patients, most with progressive disease despite prior anti-PD(L)-1 therapy and an average of 3.3 previous treatments, showed a 36% overall response rate, with two complete and 22 partial responses. The best response rate of 41% was seen in patients with progressive disease after initial anti-PD(L)-1 and anti-lymphocyte-activation gene (LAG) 3-containing regimens. Current results from the main cohort (n = 87) showed 29% of overall response rate. Combining the two cohorts, the overall response rate was 31%, indicating an effective and durable treatment option. The 12-month response rate for PD-1 heavily pretreated patients with refractory melanoma was 54%. Lifileucel provided significant responses in advanced melanoma, indicating a rapidly evolving TIL therapy, and its use as a complement for ICI antibody therapy as part of the growing immunotherapy arsenal. Accumulating data demonstrates the significant antitumor efficacy of unmodified TILs, and advances in understanding the tumor microenvironment offer opportunities to expand the therapeutic window, recognizing the true potential of TIL therapy [23].

Barras et al., investigated adoptive cell therapy using tumor-infiltrating lymphocytes (TIL-ACT), their competence of eliminating or reducing metastatic melanoma, and long-term efficacy correlation with a subset of patients. In a phase 1 clinical study, they analyzed samples from 13 metastatic melanoma patients treated with TIL-ACT. Using bulk and single-cell RNA sequencing, whole-exome sequencing, and spatial proteomic analyses of pre- and post-ACT tumor tissues, they found that TIL-ACT responders had higher baseline tumor cell-intrinsic immunogenicity and mutational burden. Compared to nonresponses, CD8 TILs in responders showed increased cytotoxicity, exhaustion, and co-stimulation, while myeloid cells exhibited enhanced type I interferon signaling. Cell–cell interaction prediction and spatial neighborhood analyses revealed that responders had rich baseline intratumoral and stromal tumor-reactive T cell networks with activated myeloid populations. Together, their findings demonstrated that TIL-ACT therapy further reprogrammed the myeloid compartment and increased TIL-myeloid networks, which is strongly associated with a response to TIL-ACT in metastatic melanoma [24].

In summary, these studies collectively underscore the significant potential of TIL therapy in treating melanoma, particularly in patients who are refractory to other treatments. Future research should continue to refine these approaches, enhancing our understanding of the tumor microenvironment and leveraging these insights to improve patient prognosis and survival rates.

## 3. Discussion

Until clinical and pathological stage III, surgery remains the cornerstone of cutaneous melanoma treatment. Immunotherapy and targeted therapies have significantly improved survival rates for patients with disseminated melanoma and those at high risk. However, despite these advancements, most of these patients succumb to melanoma, underscoring the urgent need for new therapeutic options. In this context, therapy with tumor-infiltrating lymphocytes (TILs) emerges as a promising treatment alternative for these patients. We reviewed recent medical literature from the past four years, focusing on the prognostic role of TILs in primary melanoma lesions and the therapeutic potential of TILs.

Several studies suggest a strong correlation between higher T cell infiltration in tumors and improved patient survival. Patients with significant T cell infiltration tend to have a higher rate of negative sentinel lymph nodes and better survival outcomes, even in advanced stages of melanoma [21]. This reinforces the prognostic value of TILs in melanoma.

TIL therapy has shown great promise, particularly in rescuing patients resistant to conventional immunotherapy and targeted therapies. Recently, the FDA granted accelerated approval for Lifileucel, a TIL-based therapy, based on a multicenter clinical trial where 31.5% of the 73 patients achieved an objective response according to RECIST 1.1 criteria [15]. This approval marks a significant milestone in melanoma treatment.

However, several challenges need to be addressed to advance TIL therapy. These include optimizing production time, enhancing toxicity control, improving T cell activation, and establishing a standardized platform for the rapid expansion and selection of TILs capable of effectively targeting melanoma cells. Better control of adverse effects is also desirable.

It is important to note that as this is a narrative review, there may be inherent biases related to the critical evaluation by the authors, which is a limitation of the present study. Future research should aim to address these challenges and refine TIL therapy, ultimately improving the prognosis and survival rates for advanced melanoma patients.

## 4. Summary and Outlook

Evidence strongly suggests that significant infiltration of primary melanomas by T cells is associated with a favorable prognosis for patients. In addition, immunotherapy with tumor-infiltrating lymphocytes in advanced (metastatic and/or unresectable) cutaneous melanoma has demonstrated considerable promise. However, further randomized clinical trials are necessary to fully establish its efficacy and optimize treatment protocols; however, the practical challenges such as refining TIL production, enhancing toxicity control, and establishing scalable expansion methods remain. Given the success in treating advanced melanoma, there is great potential for expanding the use of TIL therapy to other solid tumors, paving the way for broader applications in cancer treatment. Therefore, this review serves as a critical resource for future research directions, aiming to advance TIL therapy toward improving long-term outcomes and offering hope to patients facing treatment resistance.

## 5. Materials and Methods

### 5.1. Search Strategy

A search for scientific articles published between 2020 and 2024 was conducted using the terms DeCS/MeSH Tumor Interstitial Lymphocytes (Lymphocytes, Tumor-Infiltrating), Melanoma, and Neoplastic Metastasis (Neoplasm Metastasis) in the MEDLINE, LILACS and EMBASE database accessed on 

### 5.2. Article Analysis

Articles were initially analyzed based on their titles, followed by their abstracts, and then by reading the full text. Each article was reviewed by at least two researchers and summarized into a paragraph in the review results section, incorporating insights from both researchers. The introduction, objectives, discussion, and conclusion were developed collaboratively through brainstorming sessions among all researchers. These summaries formed the foundation for the final writing of the bibliographic review, with the various topics grouped and organized systematically.

### 5.3. Inclusion Criteria

The inclusion criteria were defined as articles available as free full texts from the past five years (2020 to 2024) in Portuguese, English, and Spanish, which addressed the topic of tumor-infiltrating lymphocytes in the treatment of advanced cutaneous melanoma or related to the prognostic value TILs in primary cutaneous melanoma.

### 5.4. Exclusion Criteria

The exclusion criteria were defined as articles not available as free full texts and articles outside of the period between 2020 and 2024; articles in a language other than Portuguese, English, and Spanish; articles that did not address the topics selected in the inclusion criteria.

### 5.5. Article Selection

The articles that met the inclusion criteria and did not fall under the exclusion criteria were selected. From the initial 824 references obtained from the three databases (MEDLINE, LILACS, and EMBASE), 774 were excluded for either not being relevant to the selected topics or being duplicates across databases. After reviewing the abstracts, 50 studies were deemed eligible. Out of the initial 50 studies reviewed, 36 were excluded based on the following criteria: they did not directly address the treatment or prognostic value of tumor-infiltrating lymphocytes (TILs), they focused on primary melanomas or non-cutaneous melanomas, or they evaluated the use of TILs in non-melanoma cancers. The remaining 14 publications were thoroughly re-evaluated and analyzed according to the central theme and were included in this review (Figure 3A). Among the included articles, ten were original articles, two were clinical trials, and two were reviews (Figure 3B).

## Figures and Tables

**Figure 1 ijms-25-12596-f001:**
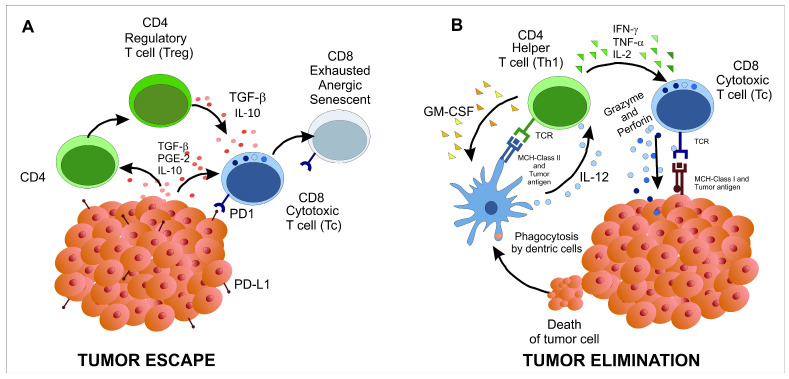
Schematic illustration of the interaction between tumor cells and immune cells. (**A**) Tumor cells promote a regulatory environment that impairs the immune system’s ability to act against them. (**B**) Immune cells overcome the tumor environment and destroy the tumor cells. The cells are represented by colors such as: light green (CD4–Th), dark green (CD4–Treg), light blue (CD8–Tc), grey (CD8 exhausted anergic or senescent). The soluble factors are represented by forms and colors such as: TGF-β, PGE-2, IL-10 (pink dots), Perforin and granzymes (blue dots), GM-CSF (orange triangles), IFN-γ, TNF-α and IL-2 (green triangles).

**Figure 2 ijms-25-12596-f002:**
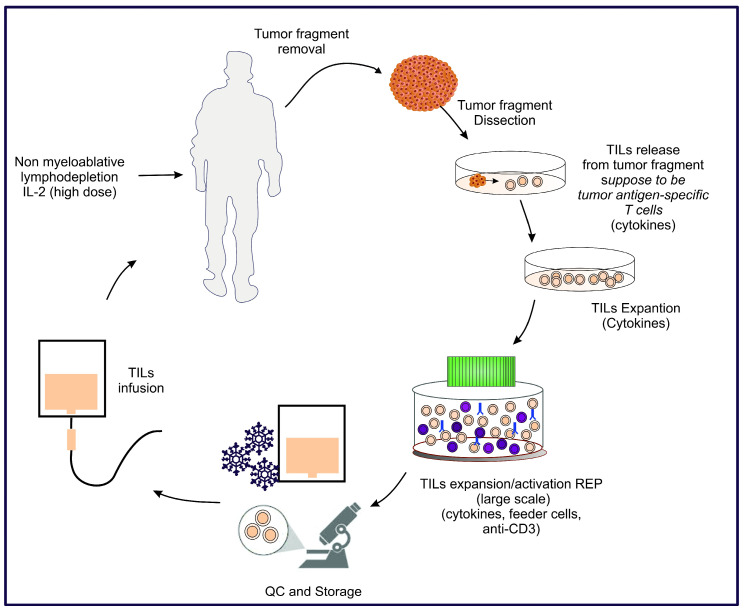
Schematic demonstration of the TIL process and therapy. The cells are represented by colors such as: light pink (TILs), brown (tumor cells), violet (feeder cells) and the stimulatory antibody was represented by blue (anti-CD3).

**Figure 3 ijms-25-12596-f003:**
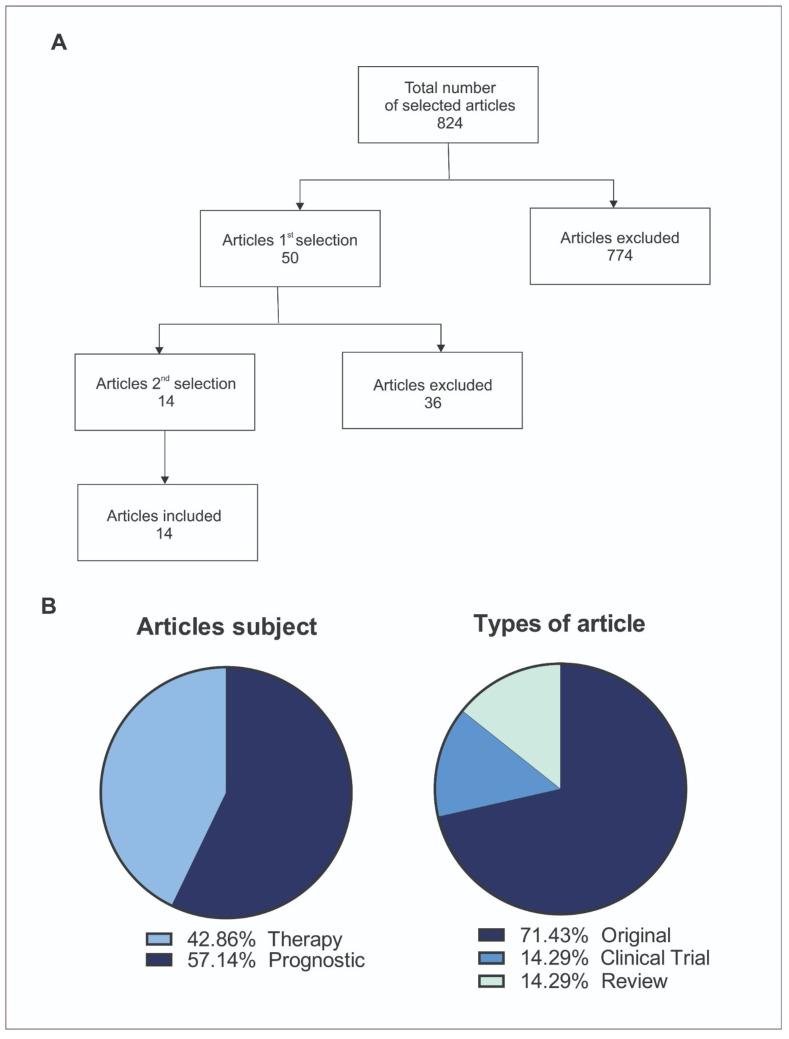
(**A**) Flowchart of articles selection. (**B**) Distribution of articles by subject and type.

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
