# Peer review of "Advances in Tumor-Infiltrating Lymphocyte (TIL) as a Prognostic Factor and for Treating Invasive Cutaneous Melanoma"

_ijms, 2024, doi:10.3390/ijms252312596_

Round 1

Reviewer 1 Report

Comments and Suggestions for Authors

The paper entitled “Tumor-Infiltrating Lymphocytes in the Treatment of Advanced Cutaneous Melanoma” presents an interesting subject being useful for development strategy in melanoma treatments, but I suggest some improvements to increase the quality of your manuscript:

11-      Please revise the title.

22-      Please add institutional address of all authors, after writing the names of institutions, departments. Also, selects a correspondence author and notes with “*” and writes the contact information such as telephone, e-mail. This information’s represents the journal request. 

33-      Please revise the abstract. Establish an objective to be correlated with entire manuscript. At ending of the abstract, please eliminates the line 19-20 “This review examines publications from the past four years focusing on TIL therapy.”

44-      Please revise keywords. Change the points between keywords with commas.

55-  In entire manuscript, the references must write in square bracket. Please correct all references from text in according with the journal request. 

66-     Please eliminate the older references, and keep references after 2004. You must chose the actualized and new references, this is the journal request.

77-      Please eliminate the lines 86-89, because did not represent interest for your manuscript development. Please consult the journal request because you wrote an article not a review.

88-   In line 92-93, 228-229, please revise the context, eliminate this sentence, “and detailed in the following text.”

99-      Please add the scientifically terms for all abbreviations, in especially for the first mention in the manuscript.

110-   Please eliminate the line 342, an entire context from line 372 to 415.

111-   In final of your manuscript, you may change Conclusions with the Summary and Outlook to describe the importance of the review.

112-   Please check the bibliography to respect the journal request and revise all.

113-   Please eliminate the Tables 1 and 2, from final of the manuscript.

Comments on the Quality of English Language

The English could be improved to more clearly express the research.

Author Response

November 11th, 2024

Dear Editor:

We are now submitting the rebuttal letter for the article “Tumor-Infiltrating Lymphocytes in the Treatment of Advanced Cutaneous Melanoma” to the
International Journal of Molecular Sciences with point-by-point response to the reviewer’s comments as follows.

Sincerely yours,

Reviewer 1

The paper entitled “Tumor-Infiltrating Lymphocytes in the Treatment of Advanced Cutaneous Melanoma” presents an interesting subject being useful for development strategy in melanoma treatments, but I suggest some improvements to increase the quality of your manuscript:

Response: Thank you for your suggestions. We have made revisions to enhance the quality of this article and ensure it is suitable for publication.

1-      Please revise the title.

Response: The title was revised and modified for “Advances in Tumor-Infiltrating Lymphocyte (TIL) as Prognostic Factor and for Treating Invasive Cutaneous Melanoma”

2-      Please add institutional address of all authors, after writing the names of institutions, departments. Also, selects a correspondence author and notes with “*” and writes the contact information such as telephone, e-mail. This information’s represents the journal request. 

Response: The corresponding authors were added and their contact information.

3-      Please revise the abstract. Establish an objective to be correlated with entire manuscript. At ending of the abstract, please eliminates the line 19-20 “This review examines publications from the past four years focusing on TIL therapy.”

Response: An objective was added to the abstract, and lines 19-20 were removed.

4-  Please revise keywords. Change the points between keywords with commas.

Response: The key words were revised

5-  In entire manuscript, the references must write in square bracket. Please correct all references from text in according with the journal request. 

Response: The entire article was revised, and brackets replaced the parenthesis.

6-     Please eliminate the older references and keep references after 2004. You must choose the actualized and new references, this is the journal request.

Response: We have eliminated the older references and replaced them with new ones.

7-      Please eliminate the lines 86-89, because did not represent interest for your manuscript development. Please consult the journal request because you wrote an article not a review.

Response: These lines were eliminated.

8-   In line 92-93, 228-229, please revise the context, eliminate this sentence, “and detailed in the following text.”

Response: These lines were removed, and an explanation was included in the following text.

9-      Please add the scientifically terms for all abbreviations, in especially for the first mention in the manuscript.

Response: All abbreviations were defined mainly in the first mention.

10-   Please eliminate the line 342, an entire context from line 372 to 415.

Response: The line 342 was eliminated.

11-   In final of your manuscript, you may change Conclusions with the Summary and Outlook to describe the importance of the review.

Response: We changed the Conclusion with the Summary and Outlook.

12-   Please check the bibliography to respect the journal request and revise all.

Response: The bibliography was revised and the numbered references within the text are now in superscript, as the journal requests.

13-   Please eliminate the Tables 1 and 2, from final of the manuscript.

Response: We have excluded Tables 1 and 2.

____________________________________________________________________________

Reviewer 2 Report

Comments and Suggestions for Authors

This article focuses on melanoma and the role of tumor-associated T-lymphocytes. It is s systematic review, but if data allows it, forest plots could be used. Additionally, figures showing how melanoma cells interact with the immune microenvironment are missing, including the mechanisms of action.

The authors may also include current clinical trials.

the authors may include a thorough description of TILs

Additional comments:

(1) Line 38. Please confirm that in Brazil the most frequent type of neoplasia is melanoma. Above breast, lung, and colorectalcancer?

(2) Line 44. In this section, please add names of the drugs.

(3) Line 72. What were histological subtype of murine tumors?

(4) Please modify the colors of Figure 2 to make it standard image.

(5) Line 404. It is stated that 770 articles were excluded. However, Figure 2 says 774. Please confirm all numbers.

(6) Please describe the properties of TILs. Are they CD8+cytotoxic T lymphocytes, Tregs, TFH cells, Th17,....? Each subpopulation have different role and regulation. I understand that this refer to cytotoxic?

(7) Line 127. Please describe the ANN in more detail.

(8) Please add figures showing how the melanoma cells interact with the immune microenvironment, including immuno-oncology markers and drugs.

(9) Is it possible to make systematic review and forest plots?

Author Response

November 11th, 2024

Dear Editor:

We are now submitting the rebuttal letter for the article “Tumor-Infiltrating Lymphocytes in the Treatment of Advanced Cutaneous Melanoma” to the
International Journal of Molecular Sciences with point-by-point response to the reviewer’s comments as follows.

Sincerely yours,

Reviewer 2

Comments and Suggestions for Authors

This article focuses on melanoma and the role of tumor-associated T-lymphocytes. It is s systematic review, but if data allows it, forest plots could be used. Additionally, figures showing how melanoma cells interact with the immune microenvironment are missing, including the mechanisms of action.

The authors may also include current clinical trials. the authors may include a thorough description of TILs

Response: Thank you for your suggestions. We have made the revisions to enhance the quality of this article and ensure it is suitable for publication.

(1) Line 38. Please confirm that in Brazil the most frequent type of neoplasia is melanoma. Above breast, lung, and colorectalcancer?

Response: We corrected the data, with non-melanoma skin cancer being the most prevalent in Brazil, above breast cancer, colorectal cancer, etc. Of these skin neoplasms, melanoma represents 3% of tumors. 

https://www.gov.br/saude/pt-br/assuntos/saude-de-a-a-z/c/cancer-de-pele#:~:text=Embora%20o%20c%C3%A2ncer%20de%20pele,das%20neoplasias%20malignas%20do%20%C3%B3rg%C3%A3o.

(2) Line 44. In this section, please add names of the drugs.

Response: We have added the names of the drugs.

(3) Line 72. What were histological subtype of murine tumors?

Response: We specified the histological subtypes of murine tumors referred (“B16 and M3 melanomas”).

(4) Please modify the colors of Figure 2 to make it a standard image.

Response: We have used now sRGB pallet of colors as you can see bellow:

(5) Line 404. It is stated that 770 articles were excluded. However, Figure 2 says 774. Please confirm all numbers.

Response: The number of excluded articles is now correct (774).

(6) Please describe the properties of TILs. Are they CD8+cytotoxic T lymphocytes, Tregs, TFH cells, Th17,....? Each subpopulation have different role and regulation. I understand that this refers to cytotoxic?

Response: We have included a description of TILs in the article as follows:

TILs were defined as a heterogeneous population of lymphoid cells, which include CD4+ helper T (Th) cells and CD8 cytotoxic T cells (Tc). Among the Th cells, Th1 supports anti-tumor immunity by secreting IFNγ, while regulatory T cells (Treg), can suppress immune responses and promote tumor tolerance. Additionally, subsets like Th17 cells may influence inflammation within the tumor and follicular helper T (Tfh) cells can help the B cells responses. Tc are the cytotoxic cells that can kill the tumor cells through degranulation of granzymes and perforin. Since these T cells are infiltrated in tumor microenvironment, they are also supposed to be mostly able to recognize tumor antigens (tumor-specific T cells) [Fridman et al, 2012].“

These lymphocytes are then cultivated and expanded in the laboratory intending to expand and activate the Tc and Th1 phenotypes tumor antigen-specific that are infiltrating the tumor, as depicted in Figure 1. “

Reference: Fridman, W., Pagès, F., Sautès-Fridman, C. et al. The immune contexture in human tumours: impact on clinical outcome. Nat Rev Cancer 12, 298–306 (2012). https://doi.org/10.1038/nrc3245.

(7) Line 127. Please describe the ANN in more detail.

Response: The ANN of U-Net was described in more detail.

(8) Please add figures showing how the melanoma cells interact with the immune microenvironment, including immuno-oncology markers and drugs.

Response: A figure showing tumor interaction with immune system was added to this article.

(9) Is it possible to make systematic review and forest plots?

Response:  The studies included in this review assess various distinct aspects related to both prognosis and therapy. Due to the diversity in outcomes and methodologies, it was not feasible to aggregate the results into a single forest plot. Each study contributes unique insights, and grouping them together might obscure these differences rather than provide a comprehensive summary. Therefore, a systematic review with descriptive analysis was prioritized over meta-analysis in this case.

Round 2

Reviewer 1 Report

Comments and Suggestions for Authors

I agree with the revision.

Reviewer 2 Report

Comments and Suggestions for Authors

Thank you for the revised version, I do not have any additional comment